# Functional Fitness Norms of Community-Dwelling Older Adults in Southern Rural Taiwan: A Cross-Sectional Study

**DOI:** 10.3390/healthcare12020213

**Published:** 2024-01-15

**Authors:** Chun-Wei Wang, Jia-Ling Yeh, Shuk-Fong Li, Chun-Ming Chen, Hsiu-Hua Wang, Cheng-Shiun He, Hsin-Tzu Lin

**Affiliations:** 1Department of Sport Performance, National Taiwan University of Sport, Taichung 404401, Taiwan; 71204001@gm.ntus.edu.tw; 2Department of Adult and Continuing Education, National Chung Cheng University, Minxiong 621301, Taiwan; d07710006@ccu.edu.tw; 3Department of Athletics Sports, National Chung Cheng University, Minxiong 621301, Taiwan; admsfl@ccu.edu.tw (S.-F.L.); jimmychen@ccu.edu.tw (C.-M.C.); grcsww@ccu.edu.tw (H.-H.W.); cshe@ccu.edu.tw (C.-S.H.)

**Keywords:** norms, functional fitness, elderly, region, Taiwan, physical activity

## Abstract

Background: Physical activity (PA) and functional fitness (FF) are crucial for promoting independent living and healthy aging in older adults. However, there is a lack of normative values for the Senior Fitness Test (SFT) among older adults (aged 55–90) living in southern rural Taiwan, particularly in the Chiayi region, which has been relatively underserved in terms of health-related resources compared to northern Taiwan. Purpose: This study aimed to determine the age- and gender-specific normative fitness scores for a large representative population of community-dwelling older adults in southern rural Taiwan. Methods: A cross-sectional descriptive study was conducted to obtain normative FF scores for 3332 community-dwelling elderly people (1057 men and 2275 women) in Chiayi, through the implementation of functional fitness tests across 72 Chiayi communities. The developed normative data served as a reference for exercise prescription for the elderly in southern Taiwan. Results: The study showed that the average normative values of all functional fitness tests declined significantly with increasing age (*p* < 0.05). Sex differences were also found in all measures of FF tests, with elderly women scoring significantly better than men on flexibility (scratch back: −1.34 ± 9.03 vs. −6.54 ± 11.07; chair sit and reach: 6.56 ± 9.44 vs. 0.56 ± 10.40) (*p* < 0.05), while men scored significantly higher than women on strength (grip strength: 30.83 ± 8.06 vs. 21.82 ± 5.32; bicep curl: 19.25 ± 5.44 vs. 17.64 ± 4.98) (*p* < 0.05). Furthermore, four FF normative scores in southern rural Taiwan were found to be significantly higher than their counterparts living in northern Taiwan. Conclusion: These findings contribute positively to the evaluation of fitness levels among older adults in southern rural Taiwan and provide a concrete reference for developing sound PA programs for this population. The results suggest that strategies aimed at promoting increased participation in PA among older adults need to consider age-, sex-, and region-specific factors.

## 1. Introduction

In the context of aging and a low birth rate, Taiwan has been experiencing a rapid increase in the proportion of elderly citizens [1]. The population of Taiwan aged 65 years and older reached approximately 16.58% in 2021 and is projected to increase to at least 20% by 2026, thereby becoming a “super-aged” society. By 2060, the elderly population is expected to account for 41% of the total population [2]. The phenomenon of population aging is a global trend that has significant implications for the health of older adults [3]. The progressive decline in the social, psychological, and physical functions of the elderly, compounded by unhealthy lifestyles, has resulted in a heightened prevalence of chronic non-communicable diseases and disability within this population [4]. In Taiwan, chronic diseases such as diabetes, heart disease, and hypertension rank among the top ten leading causes of death [5]. Consequently, the Taiwanese government is focused on promoting wellness and active aging in its aging and sedentary population. Active aging is crucial to ensuring that older adults can continue to live their lives productively and independently [6,7]. In contrast to other provinces in Taiwan, Chiayi exhibits the highest aging index and boasts the largest elderly population in the country. Consequently, prioritizing the health of the elderly emerges as a paramount concern in this particular region [8].

Physical inactivity is a significant contributor to global mortality, highlighting the importance of physical activity for improving population health [9]. Regular physical activity has been found to enhance functional fitness and promote overall health. The public health guidelines for physical activity in the United States recommend that adults engage in at least 30 min of moderate-intensity physical activity on five days each week, or at least 20 min on three days each week [10]. However, studies have revealed that a substantial proportion of elderly residents in Chiayi city, Taiwan, fail to meet these guidelines [11]. Specifically, the rate of regular exercise among rural elderly residents in Chiayi is lower compared to their urban counterparts, which may be attributed to limited access to healthcare and health-related resources [11]. Additionally, while normative functional fitness values have been established for elderly populations in northern Taiwan [12], no such reference exists for rural Taiwanese populations. Thus, to develop effective physical activity programs for the rural elderly, it is essential to first establish region-specific normative fitness performance scores. This requires comparing the physiologic attributes of elderly individuals (e.g., strength, cardiovascular function, flexibility, agility, and balance) required to perform everyday activities against the general rural population, and identifying those with fitness scores that fall below the norm. These age- and gender-specific normative fitness performance scores can be utilized by healthcare professionals and fitness instructors to design effective interventions for rural elderly populations. Despite the urgent need for such information, there is a dearth of published studies evaluating functional fitness normative scores in rural areas of Taiwan.

With the belief that “Prevention is better than Cure” and being the oldest province in Taiwan, Chiayi has increasingly emphasized the concept of healthy and localized aging. In recognition of the connection between healthy lifestyles, including regular exercise and health promotion, the Chiayi Health Bureau has adopted a preventive medicine approach rather than solely focusing on curative and therapeutic measures. Prior research has demonstrated that regular exercise among the elderly can maintain good psycho-social-physical function and reduce the prevalence of chronic diseases. Aerobic and muscular training can improve health-related physical fitness, and regular exercise among the elderly can enhance cardiorespiratory endurance, muscular strength, flexibility, balance, mobility, and body composition, thereby reducing the prevalence of chronic diseases [13]. In the aging process, there are significant differences in physical health between males and females. Previous research has found that a portion of these differences are attributed to hormonal changes that occur with age in both males and females [14,15]. Specifically, testosterone levels have demonstrated a positive correlation with muscle strength and overall physical fitness. This correlation suggests that elevated testosterone levels are linked to increased muscle strength and enhanced physical capabilities. Conversely, inadequate testosterone levels have been associated with heightened visceral fat, diminished lean body mass, and reduced muscle strength [16,17,18]. In light of the literature reviewed, the Chiayi City Health Bureau identified the imperative need to undertake a thorough evaluation of the functional health status among the elderly population in Chiayi and consequently initiated the ‘Enjoy Active Aging for Longevity and Life (EAALL)’ project. The primary objective of this research is to systematically gather relevant indicators of functional fitness among the elderly residents in Chiayi and explore gender and age differences within this context. In a comprehensive context, our research hypothesis posits that discernible variations in functional fitness among the elderly population in the Chiayi region will be revealed, contingent upon age and gender factors. Simultaneously, the project aims to establish a robust foundation for the customization of future exercise prescriptions, effectively addressing the unique needs of the elderly population in Chiayi.

## 2. Materials and Methods

### 2.1. Study Design and Participants

The study was conducted in compliance with the principles stated in the Declaration of Helsinki. This study was conducted as part of the “Active Aging for Longevity and Life (EAALL)” project implemented by the Chiayi Health Bureau. All personnel involved in this study have received training in human research ethics. Participants were required to provide written informed consent indicating their willingness to participate in the study before it began. In order to combine resources and efforts, a multidisciplinary team (the Chiayi Physical Activity and Health-Promotion Task Force) was established. This team consisted of experts from the government (Chiayi Health Bureau), academia (National Chung Cheng University), industry (Chiayi Industry Innovation and Research Center), and community (community volunteers and elderly participants). The study was conducted in two phases. The first phase involved a large-scale functional fitness screening and testing of over 3000 elderly participants (aged 55 and above) from 72 communities in Chiayi. The testing was carried out by a group of elderly volunteers who had received training from fitness experts at a nearby university. The data collected were then analyzed to develop normative functional fitness scores. The study was approved by the Health Bureau of Chiayi City, Taiwan.

### 2.2. Recruitment of Volunteers as the Functional Fitness Test Administrators

To accomplish the objective of conducting functional fitness testing on over 3000 older adults, a substantial number of test administrators and assistants were necessary. To this end, 47 dedicated and altruistic elderly volunteers were recruited from neighborhoods in Chiayi City through an open recruitment and interview process. Recognizing the importance of consistent testing procedures in ensuring accurate and interpretable data, the project director prioritized the adequate training of test administrators and volunteer assistants. The training workshop, which comprised 16 h of instruction, included topics such as the goals of the project, the fundamental concept of functional fitness, detailed descriptions of the procedures for each functional fitness test, and guidelines for providing services to elderly participants.

### 2.3. Recruitment of Participants

To provide health promotion services and functional fitness tests to a large cohort of community-dwelling older adults, a broad recruitment strategy was implemented. This involved disseminating information about the study and its objectives via multiple channels, including local TV channels, social centers, and elderly community development associations. The eligibility criteria for participants were individuals aged 55 years and older who were community-dwelling, functionally independent, ambulatory without assistive devices, and free from medical conditions or physical or cognitive limitations that would impede their ability to follow instructions to complete questionnaires and safely participate in the functional fitness tests. Participation was voluntary, and informed consent was obtained from each participant. Participants completed a Physical Activity Readiness Questionnaire (PAR-Q), a self-report containing demographic information and physical activity categories [19]. Subsequently, they engaged in a 10-min warm-up led by instructors, which included 5 min of easy aerobic exercise and 5 min of stretching major muscle groups before taking the fitness tests. The functional tests, which took approximately 20–30 min to complete, were followed by a brief counseling session that provided feedback on the individual’s fitness test report, suggestions for healthy lifestyle habits, and guidance on physical activity movements.

### 2.4. Functional Fitness Measure

To evaluate the functional fitness of elderly participants, a battery of tests previously developed and validated was employed [3]. These tests included measures of body composition via body mass index (BMI), body flexibility via the back scratch and chair sit and reach, upper- and lower-muscular strength via 30 sec arm curls and 30 s chair stand, dynamic balance via timed up-and-go test, and cardiovascular endurance via the 2 min step test. In addition, the handgrip test was employed to assess forearm strength, and the clinical test of sensory interaction on balance (CTSIB) was used to measure static balance. The CTSIB balance test has been shown to be a reliable tool for assessing how well healthy older adults use sensory inputs, such as vision, somatosensory, and vestibular, to maintain balance [20].

The methods for functional fitness test were as follows:(1)BMI:

BMI was calculated by converting the participant’s measured height and weight using the formula: BMI = Weight (kg)/Height^2^ (m^2^).

(2)Scratch Back:

Participants touched hands behind their back, freely choosing the softer hand as the over-the-shoulder side. The distance between the middle fingers was measured, with positive values indicating an overlap and negative values indicating no contact. After two practice attempts, participants chose the hand with the better result for the formal test, conducted twice, and the superior score was selected.

(3)Chair Sit and Reach:

Participants sat on the edge of a chair, with one knee bent at a 90-degree angle flat on the floor and the other leg extended forward on a seat-edge flexometer. With hands overlapped and pushing forward on the sensor, data were measured and displayed. The formal test was conducted twice, and the superior score was selected.

(4)Grip Strength:

Standing upright with arms and forearms extended and relaxed, shoulders inward, participants squeezed tightly while keeping the body fixed. The dominant hand was measured, and two measurement values were recorded, with the maximum value selected.

(5)Bicep Curl:

Participants performed as many single-arm bicep curls as possible within 30 s using the correct posture. Female participants used a 5-pound dumbbell, and male participants use an 8-pound dumbbell. The assessment was based on the total number of repetitions completed within 30 s.

(6)Chair Stand:

Participants sat on a 0.43-m-high chair, arms crossed over the chest, and were required to perform as many stand-up movements as possible within 30 s for assessment. Scoring was based on the total number of stands completed within 30 s.

(7)The 2.44 Up-and-Go:

Seated halfway on a chair, participants, upon hearing instructions, stood up, walked to a cone located 2.44 m away, circled the cone, and walked back to sit on the chair. Scoring was based on the time taken to complete the test, measured in seconds.

(8)The 2 min Step Test:

Participants stood on a stepper, starting with the right foot, and stepped in place for two minutes. Each step was counted when the foot reached the midpoint between the knee and hip joint. Both knees had to reach this height for a count to be registered.

(9)Eye Opened and Eye Closed:

This study utilized the Clinical Test of Sensory Interaction on Balance (CTSIB) embedded in the Biodex Portable Balance System (Biodex Medical System, Shirley, NY, USA) to assess participants’ static balance ability. The test sequence included four modes: 1. Eyes open on a stable surface, 2. Eyes closed on a stable surface, 3. Eyes open on an unstable surface, and 4. Eyes closed on an unstable surface. Each mode had a testing duration of 30 s, with a 20-s rest period between modes. The primary assessment parameter in this study was the Sway Index (Si), calculated as the standard deviation in the stability index. Si represents the participants’ sway angle, with values ranging from 0.1 to 4. A higher index indicates greater instability during the testing process.

### 2.5. Data Analyses

The Statistical Package for Social Studies (SPSS v21) was utilized for data analysis. The participants were categorized into eight groups based on their age in 5-year intervals: 55–59, 60–64, 65–69, 70–74, 75–79, 80–84, 85–90, and 90 years or older. Analyses were carried out separately for both men and women. Descriptive statistics were calculated, including means, standard deviations, frequencies, and percentages for categorical demographic variables such as level of education, exercise habits, type of disease, and smoking habit. A five-rank percentile (10th, 30th, 50th, 70th, and 90th) was calculated to construct the percentile grid. Independent samples *t*-tests were used to compare gender differences within the same age group for each functional fitness parameter. A one-way ANOVA was then used to compare each of the functional test parameters of the different age groups against the 55–59 age group. The statistical significance level was set at *p* < 0.05.

## 3. Results

The data collection of functional fitness tests for 3332 older adults from 72 neighborhoods in Chiayi was successfully completed. A total of 1656 participants from 37 neighborhoods completed the FF tests in 2013, while 1676 participants from the other 35 neighborhoods were recruited and tested in 2014. Table 1 presents the general demographic characteristics of the elderly participants. The data indicate that there were twice as many women (68.3%) as men (31.7%) in the study. Only 17% of the participants were aged 80 and older, and almost half (48.7%) of the subjects received no education or elementary education. Furthermore, men had a higher educational level than women. Overall, the participants were generally active, with over 70% of both genders reporting that they exercised regularly at least three times per week. Furthermore, 98.8% of females reported no smoking, while their counterparts reported 89.8%. However, 49% of all participants reported having one or more chronic diseases, and one-third (32.6%) of all participants had high blood pressure. The demographic characteristics of the elderly participants are presented in Table 1.

### 3.1. Comparison of BMI, Functional Fitness, and Balance among Different Age Groups

In this study, we compared the means and standard deviations of BMI and functional fitness (FF) test variables across seven age groups to those of the 55–59 group (middle-age). Table 2 presents the results of BMI and FF tests stratified by age and gender. The ANOVA analysis revealed significant F ratios on all BMI and FF measures (*p* < 0.001) when comparing the seven age groups (60–64; 65–69; 70–74; 75–79; 80–84; 85–89; 90 or older) to the 55–59 group. These results indicate that significant differences were found on most FF variables across the majority of the age groups, and gradual declines in all functional fitness test scores were observed over almost all the 5-year spans for both men and women. We did not find a significant difference (*p* > 0.05) in BMI from the ages of 55 to 84, but we found lower BMI for the 85-or-above age group in male elderly participants. However, we did not find any significant differences (*p* > 0.05) among all the age ranges for elderly women.

### 3.2. Mean Differences in Functional Fitness, and Balance between Male and Female

In accordance with the data presented in Table 3, significant differences were observed between elderly men and women in terms of their performance on various functional fitness tests. Specifically, elderly women demonstrated significantly better performance (*p* < 0.05) on both upper-body flexibility (back scratch) and lower-body flexibility (chair sit and reach) compared to elderly men. Conversely, elderly men exhibited significantly (*p* < 0.05) higher scores than women in other functional fitness tests, including grip strength, upper-body strength measured by 30-sec arm curl, and lower-body strength measured by 30-s chair stand. No significant differences were found between elderly men and women on measures of BMI, static balance, cardiovascular function measured by the 2 min step test, and dynamic balance as measured by the 2.44 M up-and-go test.

### 3.3. Functional Fitness Normative Scores Established for Rural Chiayi Elderly

Table 4 displays the score equivalents for men and women across all age groups on each functional fitness test item at the 10th, 30th, 50th, 70th, and 90th percentiles. These percentiles allow for the determination of an individual’s functional fitness relative to their age group and gender. Scores falling between the 30th and 70th percentiles within an age range are considered normal, representing the middle 40% of a distribution. Scores below the 30th percentile are below average, while scores above the 70th percentile are above average. The normative scores presented in Table 4 provide sex- and age-specific percentile values (P10, P30, P50, P70, P90) for eight functional fitness measures, including body mass index, upper- and lower-body flexibility, grip strength, balance, upper- and lower-body strength, dynamic balance and agility, and cardiovascular function.

## 4. Discussion

Normative scores are essential for identifying individuals at risk and for evaluating fitness programs [12]. The primary objective of this study was to develop sex- and age-specific normative values for functional fitness tests for community-dwelling older adults in a southern rural county with limited healthcare resources and information on healthy lifestyles, such as exercise and nutrition, compared to northern urban areas [21]. The study established normative functional fitness scores for 3332 older adults aged 55 to 90 or older. The most significant finding of this study is the consistent and progressive age-specific decline in performance on all functional fitness test variables for both men and women, which aligns with the previous literature demonstrating that functional fitness indicators of the elderly decline with age for both genders [12,22,23,24,25,26,27]. Demura et al. [28] conducted a study on 75-year-olds and confirmed that physical fitness declines with age. Aging-associated degradation was found in all fitness testing parameters, especially in flexibility, balance, and agility [24]. Ho et al. [29] found that age and gender significantly predicted flexibility and physical power measured by arm curl, chair stand, the 2.44 m up-and-go, and the 2 min step test.

BMI is an indicator of body composition often used to determine whether an individual is overweight or obese. However, in the elderly, a decrease in BMI is typically caused by the loss of muscle mass, bone, and other soft tissue rather than the loss of body fat [30]. Therefore, maintaining BMI at a healthy level with more muscular mass than fat mass is essential. The result of no difference found in BMI between the age group of 55–60 to 80–84 and finding lower BMI in the 85-or-above age group for male elderly is consistent with reports from other studies [23,31]. In this study, the highest BMI recorded in men aged 65–69 years was (24.88 ± 2.88), and the lowest BMI was found in the oldest subjects aged above 90 years or above (22.18 ± 3.68). Elderly women had a very stable BMI recorded among all age groups, with females of 85–89 age group having the lowest BMI (23.54 ± 3.99), while the 75–79 age group had the highest BMI (24.83 ± 3.50). This result is predictable because older women in rural Chiayi tend to be more socially active than men, who typically lead sedentary lives at home, resulting in a loss of muscle mass and strength. This is also a reason why we recruited two-thirds women and only one-third men out of the 3332 participants in this study.

This study reveals that there are comparable findings in terms of the overall degeneration rate of functional fitness and balance across genders among various age groups. Among male elderly individuals, general physical deterioration typically starts around the age of 70, with most FF performances, including upper-body flexibility, lower-limb strength, eye-closed balance, and cardiorespiratory endurance, being lower than those of elderly individuals aged 55–69 [23]. However, agility degeneration occurs in the age group of 65–69, and forearm strength (grip test) and upper-limb strength (30-s arm curl) show deterioration even earlier in the 60–64 age group when compared to middle-aged males (55–59). For females, it is noteworthy that apart from eye-open balance, upper-limb flexibility, and cardiorespiratory endurance, four FF parameters, namely, forearm strength, upper- and lower-limb strength, and eye-closed balance, start declining as early as the age group of 60–69. Consequently, physical educators, and health and fitness instructors from government and private sectors, such as gym businesses, should promote health and prescribe exercise programs to younger elderly participants [31]. The normative values for all functional fitness measurements differ between women and men, with men exhibiting better muscle strength parameters, including grip strength, upper- and lower-body strength, and upper- and lower-body flexibility. This finding is consistent with previous evidence regarding the decline in functional fitness with advancing age [24,25]. Aging data in terms of gender show that the decrease in muscle mass with age, along with a decrease in muscle density and an increase in inter-muscular fat mass, is most prominent in women [32]. In terms of gender’s effect on functional fitness, elderly men were found to be superior to women in terms of muscular endurance, possessing more muscle mass, better grip strength and functional mobility, and higher levels of physical activity than elderly women [33]. Furthermore, the flexibility of the elderly is a frequently discussed topic, with age-related decreases being well-known. Joint mobility and upper- and lower-limb flexibility in elderly women are better than those in elderly men [34]. Hence, promoting physical activity and lifestyle interventions can reduce functional incapability among elderly individuals. Therefore, the intensity of physical activity should consider muscle fitness, dynamic balance/agility, and maintaining or increasing joint motion.

This study examined the impact of geographic location on FF and BMI among elderly individuals in Taiwan. The establishment of FF norms for elderly individuals in Taiwan revealed differences in FF normative scores and BMI between community-dwelling elderly individuals in northern and southern Taiwan. Specifically, elderly individuals in southern Taiwan had higher FF normative scores in forearm strength, lower-body strength, upper-body flexibility, and cardiovascular capability, as well as lower BMIs, when compared to their northern counterparts [12]. The study also explored differences between elderly individuals living in rural and urban areas. Deluga et al. [35] found that the healthy lifestyles of elderly individuals were influenced by several factors, including age, gender, health status, level of education, and place of residence. Participants from rural areas tended to have lower scores in physical activity and nutrition, while other studies found that the residents of rural areas had healthier lifestyles in terms of physical activity and diet than those living in cities [36]. Additionally, our study revealed that the participants were predominantly female (N = 2275, 68.3%), which differs from previous research that recruited more male subjects (54.7%) than female elderly (45.3%) in northern urban Taiwan [12]. Moreover, participants in northern Taiwan had a significantly higher level of education (28%) than those in southern Taiwan (16.1%). Educational attainment was also associated with healthy behaviors such as appropriate physical activity levels [35]. These demographic differences highlight the importance of considering cultural and lifestyle factors when designing exercise and health promotion programs for elderly individuals. Finally, Tang et al. [10] found that the rate of regular exercise and multiple chronic diseases was higher among urban elderly residents than rural elderly residents in Chiayi. To address these disparities, health authorities should promote exercise behavior and functional fitness programs that enhance muscle strength, agility, and balance, specifically targeting elderly individuals living in rural areas.

From the perspective of preventing disability, it is important to emphasize the development of muscle strength in the elderly, particularly among elderly women, and encourage them to maintain their ability to age actively. In summary, the data presented in this study offer information on normal rates of change across age groups. This database not only provides normative functional fitness scores specifically for elderly citizens, but also enables comparison that can help identify areas of weakness in individuals and plan exercise health-promotion interventions that can prevent or reduce physical frailty and disability. Moreover, by applying the concepts of “localized aging” and “elderly as valuable social assets rather than social burdens,” the majority of the volunteering assistants who administered the functional fitness tests were elderly individuals from the nearby community. According to a study, volunteering assistance is an important aging policy in Taiwan, and the goal of improving functional fitness, promoting societal value, and encouraging social participation among the elderly has been achieved [7]. In addition, the comprehensive findings from previous research indicate a widespread decline in functional fitness among older individuals across different countries globally, particularly after the ages of 80 and 85 [26,27,37]. These studies further highlight that women tend to outperform men in balance, while men exhibit relative advantages in muscular strength and aerobic capacity [26,27,37]. Our study results align with these international findings, providing additional confirmation of this common aging phenomenon. The consistency of these findings underscores the clinical significance of developing preventive and educational programs tailored to older adults. Implementing such programs can empower individuals to adopt proactive lifestyles, contributing to successful aging.

### Strengths and Limitations

This study represents the first attempt to establish functional fitness normative scores in rural elderly individuals residing in southern Taiwan, utilizing the Senior Functional Test battery. The resulting normative-referenced values for functional fitness in southern Taiwan have the potential to be utilized for public health and clinical practice screening, exercise prescription and tracking, and individual self-assessment of functional fitness status. Furthermore, this study only includes participants from a specific region in southern Taiwan, and its findings should not be extrapolated to other geographical areas or ethnic groups. One potential limitation of this study is its use of a cross-sectional design, which precludes the establishment of causal relationships. Furthermore, data collection for this study took place in 2013–2014, limiting our insight into the current functional fitness status among the elderly in the Chiayi region. However, we consider it crucial to conduct a 10- or 20-year follow-up to capture changes over the long term. Additionally, the study cohort consisted of twice as many women as men, and only 17% of participants were aged 80 years or older. Given that older individuals and men are at increased risk of developing chronic diseases [38], it is imperative to include these populations in future physical activity and health promotion campaigns.

## 5. Conclusions

The functional fitness performance of individuals aged 65–69 or younger showed a consistent trend, but the muscular strength and agility/dynamic balance decreased rapidly after the age of 70.The functional fitness scores for both men and women decreased with age, with men exhibiting better muscular strength in the forearm, upper body, and lower body, while women had better flexibility in the upper and lower limbs, as well as better static balance.Elderly individuals residing in southern rural Taiwan had relatively higher functional fitness scores for lower-body strength, upper-body flexibility, cardiovascular function, and grip strength when compared to their northern counterparts. Government institutions and social organizations aimed at improving the quality of aging should consider local differences between rural and urban senior communities and tailor their actions accordingly.This study suggests that although the rural elderly community reported regular exercise habits, they still need to enhance their exercise behavior based on the results of the functional fitness test in order to effectively achieve the recommended exercise and improve their functional fitness, particularly in terms of muscle strength and dynamic balance.

## Figures and Tables

**Table 1 healthcare-12-00213-t001:** Demographic characteristics of the participants (n = 3332).

Characteristics	No. of Participants (Percent %)		
Gender	Male (%) 1057 (31.7%)	Female (%) 2275 (68.3%)	Total (%) 3332 (100%)
Age Interval			
55–59	79 (7.4%)	349 (15.3%)	428 (12.9%)
60–64	152 (14.2%)	407 (18%)	559 (16.7%)
65–69	195 (18.5%)	415 (18.2%)	610 (18.4%)
70–74	227 (21.6%)	440 (19.2%)	667 (19.9%)
75–79	170 (16.1%)	342 (15.1%)	512 (15.4%)
80–84	137 (12.9%)	213 (9.3%)	350 (10.6%)
85–89	72 (6.7%)	81 (3.4%)	153 (4.7%)
90 or above	25 (2.5%)	28 (1.2%)	53 (1.7%)
Education			
No Edu.	96 (9.1%)	496 (21.8%)	592 (17.8%)
Primary	282 (26.7%)	746 (32.8%)	1028 (30.9%)
Junior High	129 (12.2%)	341 (15.0%)	470 (14.1%)
Senior High	262 (24.8%)	441 (19.4%)	703 (21.1%)
University	266 (25.2%)	233 (10.2%)	499 (14.9%)
Graduate	21 (2.0%)	18 (0.8%)	39 (1.2%)
Regular Exercise			
Yes	803 (76.0%)	1643 (72.2%)	2446 (73.4%)
No	254 (24.0%)	632 (27.8%)	866 (26.6%)
Exercise Frequency			
No exercise	98 (9.3%)	242 (10.7%)	340 (10.2%)
1 time/week	36 (3.4%)	127 (5.6%)	163 (4.9%)
2 times/week	56 (5.3%)	125 (5.5%)	181 (5.4%)
3 times/week	867 (82.0%)	1781 (78.3%)	2648 (79.5%)
Type of Disease			
No	492 (46.5%)	1206 (53.0%)	1698 (51%)
Blood Pressure	373 (35.3%)	712 (31.3%)	1085 (32.6%)
Diabetes	91 (8.6%)	171 (7.5%)	262 (7.9%)
Heart Disease	57 (5.4%)	100 (4.4%)	157 (4.7%)
Asthma	11 (1.0%)	23 (1.0%)	34 (1%)
others	33 (3.0%)	63 (2.7%)	96 (2.8%)
Smoking			
No	949 (89.8%)	2248 (98.8%)	3197 (95.9%)
Occasion	27 (2.6%)	14 (0.6%)	41 (1.2%)
Less than 10/day	32 (3.0%)	9 (0.4%)	41 (1.2%)
10~20/day	33 (3.1%)	4 (0.2%)	37 (1.1%)
Over 20/day	16 (1.5%)	0(0.0%)	16 (0.6%)

**Table 2 healthcare-12-00213-t002:** Comparison of BMI, functional fitness, and balance scores among different age groups (n = 3332).

Test Items	55~59	60~64	65~69	70~74	75~79	80~84	85~89	90 or Older
Male								
BMI	24.86 ± 3.91	24.58 ± 3.22	24.88 ± 2.88	24.56 ± 3.20	24.18 ± 3.17	24.10 ± 2.90	23.15 ± 2.45 *	22.18 ± 3.68 *
Scratch Back	−3.53 ± 10.85	−3.87 ± 10.50	−3.36 ± 9.30	−7.42 ± 11.33 *	−7.40 ± 10.85 *	−10.19 ± 11.85 *	−12.07 ± 11.09 *	−7.29 ± 12.15 *
Chair Sit and Reach	1.96 ± 10.71	4.30 ± 9.87	2.63 ± 9.97	2.02 ± 10.17	−0.68 ± 8.90	−3.29 ± 10.33 *	−5.05 ± 9.76 *	−10.04 ± 10.47 *
Grip Strength	36.03 ± 8.28	33.80 ± 8.53 *	34.36 ± 7.28 *	31.30 ± 6.43 *	29.11 ± 6.99 *	26.24 ± 6.25 *	24.48 ± 6.05 *	19.61 ± 6.56 *
Bicep Curl	22.95 ± 6.24	21.24 ± 5.36 *	21.11 ± 4.9 *	19.41 ± 4.32 *	17.72 ± 5.04 *	17.08 ± 4.90 *	14.99 ± 4.35 *	14.20 ± 5.82 *
Chair Stand	20.57 ± 5.85	19.72 ± 5.93	19.67 ± 5.62	17.71 ± 5.12 *	16.63 ± 5.55 *	14.68 ± 5.55 *	11.86 ± 4.52 *	10.52 ± 4.46 *
The 2.44 Up-and-Go	4.88 ± 1.21	5.35 ± 1.20	5.51 ± 1.23 *	6.08 ± 1.94 *	7.04 ± 3.33 *	7.79 ± 3.33 *	8.06 ± 2.34 *	9.16 ± 4.12 *
The 2 min Step Test	107.95 ± 18.69	107.55 ± 18.68	107.08 ± 18.65	100.07 ± 21.09 *	95.80 ± 21.35 *	94.78 ± 21.33 *	86.61 ± 26.60 *	70.45 ± 27.74 *
Eye opened	0.53 ± 0.42	0.46 ± 0.14	0.51 ± 0.24	0.55 ± 0.28	0.58 ± 0.29	0.56 ± 0.17	0.63 ± 0.22 *	0.65 ± 0.22 *
Eye closed	1.68 ± 0.64	1.77 ± 0.84	1.82 ± 0.78	1.90 ± 0.56 *	2.16 ± 0.65 *	2.25 ± 0.69 *	2.43 ± 1.15 *	2.22 ± 0.66 *
Female								
BMI	24.10 ± 3.79	24.33 ± 3.25	24.51 ± 3.86	24.57 ± 3.17	24.83 ± 3.50 *#	24.28 ± 3.68	23.54 ± 3.99	24.01 ± 4.33
Scratch Back	1.69 ± 7.71 #	1.07 ± 7.70 #	−1.12 ± 8.83 *#	−1.2 ± 8.29 *#	−2.78 ± 9.32 *#	−5.52 ± 10.53 *#	−8.65 ± 10.18 *#	−9.27 ± 10.56 *
Chair Sit and Reach	8.57 ± 10.32 #	8.94 ± 9.47 #	7.99 ± 8.88 #	5.97 ± 8.49 *#	5.34 ± 8.37 *#	2.57 ± 9.20 *#	−0.79 ± 10.39 *#	1.46 ± 7.62 *#
Grip Strength	25.18 ± 4.83 #	23.76 ± 4.73 *#	22.72 ± 4.63 *#	21.63 ± 4.51 *#	19.76 ± 4.55 *#	17.73 ± 4.75 *#	15.96 ± 4.68 *#	14.69 ± 5.73 *#
Bicep Curl	19.84 ± 5.03 #	18.97 ± 4.42 *#	18.27 ± 4.31 *#	17.54 ± 4.58 *#	16.96 ± 4.60 *	14.17 ± 4.87 *#	13.38 ± 4.78 *#	10.52 ± 5.67 *#
Chair Stand	20.15 ± 6.26	18.56 ± 5.10 *#	17.51 ± 4.76 *#	16.30 ± 4.69 *#	15.38 ± 5.39 *#	12.22 ± 5.23 *#	10.76 ± 4.62 *	10.20 ± 5.84 *
The 2.44 Up-and-Go	5.29 ± 1.04 #	5.50 ± 1.17	6.27 ± 5.72 *#	6.48 ± 1.47 *#	7.49 ± 4.97 *	8.60 ± 3.55 *#	9.74 ± 5.77 *#	8.45 ± 3.06 *
The 2 min Step Test	108.00 ± 17.33	107.58 ± 16.96	106.18 ± 50.94	98.07 ± 21.39 *	95.20 ± 22.64 *	83.80 ± 26.01 *#	84.59 ± 24.77 *	68.77 ± 27.47 *
Eye opened	0.50 ± 0.51	0.48 ± 0.18	0.52 ± 0.25	0.55 ± 0.30 *	0.57 ± 0.29 *	0.62 ± 0.49 *	0.61 ± 0.23 *	0.73 ± 0.50 *
Eye closed	1.71 ± 0.75	1.85 ± 0.83 *	1.94 ± 0.95 *	2.00 ± 0.55 *#	2.06 ± 0.73 *	2.21 ± 0.65 *	2.33 ± 0.81 *	2.22 ± 0.89 *

* Significant difference from 55 to 59, *p* < 0.05. # Significant difference between male and female, *p* < 0.05.

**Table 3 healthcare-12-00213-t003:** Mean differences in BMI, functional fitness, and balance between male and female (n = 3332).

	Male		Female		Mean Difference	T
	Mean	SD	Mean	SD		
Age	72.07	8.83	69.34	8.78	2.727	8.328 ***
Height	164.03	6.32	153.60	5.73	10.430	45.659 ***
Weight	65.80	10.30	57.67	8.96	8.127	22.057 ***
BMI	24.37	3.17	24.41	3.56	−0.043	−0.349
Scratch Back	−6.54	11.07	−1.34	9.03	−5.198	−13.338 ***
Chair Sit and Reach	0.56	10.40	6.56	9.44	−5.996	−15.931 ***
Grip Strength	30.83	8.06	21.82	5.32	9.014	33.158 ***
Bicep Curl	19.25	5.44	17.64	4.98	1.612	8.169 ***
Chair Stand	17.44	6.028	16.72	5.78	0.720	3.247 ***
The 2.44 Up-and-Go	6.36	2.55	6.58	3.77	−0.213	−1.663
The 2 min Step Test	100.03	21.91	100.13	29.77	−0.096	−0.094
Eye opened Bal	0.54	0.26	0.54	0.34	0.003	0.287
Eye closed Bal	1.98	0.75	1.96	0.77	0.020	0.707

Significant difference between male and female, *** *p* < 0.001.

**Table 4 healthcare-12-00213-t004:** The normative scores of sex- and age-specific percentile values (P10, P30, P50, P70, P90) for BMI and eight functional fitness.

	Male (n = 1057)					Female (n = 2275)				
	10%	30%	50%	70%	90%	10%	30%	50%	70%	90%
Body Mass Index (BMI)										
55–59	20.26	23.46	24.55	26.05	28.73	20.03	21.82	23.49	25.58	29.15
60–64	20.35	22.75	24.76	26.21	28.73	20.56	22.56	23.98	25.70	28.34
65–69	21.35	23.02	24.84	26.34	28.80	20.10	22.42	24.10	26.03	29.33
70–74	20.62	23.18	24.55	25.85	28.81	20.69	22.87	24.26	25.89	28.83
75–79	20.00	22.45	24.26	26.00	28.08	20.56	23.22	24.66	26.28	29.50
80–84	20.89	22.7	23.81	25.00	28.19	20.26	22.00	23.73	26.00	29.29
85–89	20.32	21.88	23.10	24.46	26.16	19.11	21.52	23.46	25.05	27.79
90–95	17.88	20.45	22.07	24.04	26.59	18.19	22.03	24.04	25.96	28.93
Scratch Back										
55–59	−16.40	−6.20	−1.00	3.00	8.70	−7.00	0.00	3.00	5.50	9.00
60–64	−19.90	−8.70	−1.00	3.00	7.00	−8.00	0.00	3.00	5.00	8.30
65–69	−17.00	−6.05	−1.00	2.00	6.70	−13.00	−4.00	1.00	4.00	8.00
70–74	−24.00	−14.00	−5.50	1.00	5.00	−12.00	−3.68	1.00	3.00	7.00
75–79	−22.00	−14.00	−7.00	1.00	5.00	−15.00	−6.05	0.00	3.00	7.00
80–84	−28.00	−17.00	−10.00	0.00	4.00	−19.00	−10.00	−3.46	1.00	6
85–89	−28.70	−17.70	−13.00	−4.00	2.00	−23.30	−14.00	−6.50	−3.00	3.30
90–95	−26.33	−13.30	−3.8	3.00	5.00	−23.50	−13.50	−9.50	−2.00	3.50
Chair Sit and Reach										
55–59	−10.60	−1.00	1.00	7.00	16.00	−4.00	3.00	9.00	14.09	21.52
60–64	−9.00	0.24	4.00	9.00	16.00	0.00	4.00	9.00	14.00	20.00
65–69	−10.00	−1.55	2.50	7.00	14.70	−1.00	3.00	7.00	13.00	19.00
70–74	−10.92	−2.00	2.00	7.00	15.60	−2.00	2.00	5.00	10.00	17.00
75–79	−14.20	−2.00	1.00	4.00	10.00	−3.00	2.00	4.26	9.00	16.00
80–84	−18.00	−8.20	0.00	2.00	8.40	−9.10	0.00	3.00	6.00	13.00
85–89	−20.00	−10.70	0.00	2.00	4.90	−15.30	−1.00	0.00	3.00	9.30
90–95	−24.20	−17.80	−10.00	−3.00	2.60	−4.50	0.00	1.00	3.46	10.50
Grip Strength										
55–59	25.17	30.98	37.40	40.30	45.90	19.70	22.90	24.80	27.55	30.90
60–64	22.01	30.72	34.90	38.74	43.09	17.66	21.40	23.90	26.10	29.20
65–69	25.06	29.94	35.00	38.40	44.32	17.36	20.49	22.70	24.90	28.57
70–74	23.16	27.90	31.40	34.90	39.40	16.10	19.31	21.65	23.80	27.10
75–79	19.99	25.57	29.00	33.55	38.02	13.78	17.47	19.80	22.10	24.91
80–84	18.42	23.78	26.30	29.34	33.54	11.89	15.80	17.90	19.73	22.41
85–89	17.52	22.96	24.70	26.10	30.97	9.6	13.70	16.50	18.30	22.30
90–95	14.06	16.38	18.30	23.08	27.72	9.10	12.88	14.00	15.08	19.24
Bicep Curl										
55–59	16.00	20.00	23.00	25.00	30.00	14.00	17.00	20.00	22.00	26.00
60–64	15.00	18.00	20.00	24.00	28.00	13.00	17.00	19.00	21.00	25.00
65–69	15.00	18.00	21.00	23.00	27.60	13.00	16.00	18.00	20.00	24.00
70–74	14.60	17.00	19.00	21.20	24.00	12.00	15.00	18.00	20.00	23.70
75–79	12.00	15.00	17.00	20.00	24.00	12.00	15.00	17.00	19.00	23.00
80–84	11.00	15.00	16.50	19.50	23.50	8.00	11.00	15.00	17.00	20.40
85–89	10.00	13.30	15.00	17.00	20.00	6.60	11.00	14.00	16.20	18.00
90–95	8.00	11.20	14.00	16.80	21.20	4.00	6.80	10.00	12.20	19.40
Chair Stand										
55–59	14.00	18.00	20.00	24.00	28.20	13.70	17.00	19.00	22.00	28.00
60–64	12.90	16.00	19.00	23.00	28.10	13.00	16.00	18.00	21.00	26.00
65–69	13.00	16.00	19.00	22.00	27.00	12.00	15.00	17.00	19.00	23.00
70–74	12.00	15.00	17.00	20.00	24.50	11.00	14.00	16.00	18.00	22.00
75–79	10.00	13.70	16.00	20.00	23.00	10.00	12.00	15.00	17.00	22.00
80–84	8.00	12.00	14.00	17.00	22.00	6.00	10.00	12.00	14.90	18.00
85–89	5.00	10.00	12.00	14.00	18.00	6.00	8.10	10.50	12.00	16.00
90–95	5.00	7.40	11.00	12.00	15.80	3.00	7.20	10.00	11.00	16.20
The 2.44 Up-and-Go										
55–59	3.78	4.30	4.71	5.00	5.74	4.13	4.71	5.11	5.63	6.52
60–64	4.00	4.84	5.00	5.60	6.49	4.35	4.90	5.32	5.84	6.72
65–69	4.09	4.81	5.34	5.99	7.00	4.70	5.32	5.78	6.26	7.30
70–74	4.48	5.37	5.94	6.44	7.37	5.00	5.72	6.24	6.94	8.00
75–79	5.00	5.86	6.32	7.00	9.02	5.44	6.20	6.81	7.43	9.23
80–84	5.23	6.31	7.00	7.86	10.66	6.03	6.75	7.60	8.81	11.71
85–89	6.00	6.76	7.32	8.86	10.67	6.18	6.49	7.63	9.42	17.63
90–95	6.26	6.47	7.99	9.06	14.58	6.21	6.62	7.19	8.70	12.82
The 2 min Step Test										
55–59	89.40	102.00	110.00	116.90	127.00	90.20	101.00	109.00	116.00	125.80
60–64	89.00	100.00	108.50	116.00	127.30	88.40	101.00	108.00	115.00	126.60
65–69	87.00	98.00	106.00	114.00	128.70	83.00	97.00	105.00	113.00	125.00
70–74	79.10	94.00	103.00	111.00	121.90	70.00	92.00	101.00	108.00	120.30
75–79	70.60	90.90	97.00	107.00	120.00	68.60	90.00	100.00	108.00	118.00
80–84	72.10	89.00	99.00	103.70	117.90	44.60	77.00	89.00	100.00	111.80
85–89	41.80	80.00	94.00	102.20	113.00	50.60	73.80	89.50	101.90	110.00
90–95	30.50	60.00	80.00	90.70	99.80	30.00	52.30	79.00	88.70	97.70
Eye opened										
55–59	0.30	0.35	0.42	0.53	0.70	0.28	0.36	0.42	0.52	0.71
60–64	0.30	0.37	0.44	0.53	0.61	0.30	0.39	0.45	0.54	0.70
65–69	0.31	0.39	0.47	0.57	0.76	0.32	0.40	0.47	0.57	0.75
70–74	0.33	0.43	0.51	0.59	0.81	0.33	0.42	0.49	0.58	0.77
75–79	0.35	0.43	0.52	0.59	0.82	0.35	0.44	0.52	0.60	0.80
80–84	0.35	0.46	0.55	0.64	0.77	0.37	0.47	0.53	0.62	0.82
85–89	0.42	0.52	0.59	0.68	0.84	0.39	0.49	0.57	0.64	0.86
90–95	0.45	0.56	0.64	0.70	0.81	0.44	0.52	0.58	0.75	0.86
Eye closed										
55–59	1.16	1.38	1.57	1.74	2.22	1.08	1.33	1.59	1.87	2.34
60–64	1.21	1.47	1.69	1.92	2.27	1.16	1.48	1.73	2.04	2.60
65–69	1.21	1.44	1.75	2.00	2.44	1.28	1.58	1.86	2.08	2.54
70–74	1.19	1.58	1.90	2.16	2.64	1.33	1.70	1.97	2.23	2.72
75–79	1.46	1.81	2.07	2.38	2.99	1.35	1.72	1.96	2.28	2.83
80–84	1.46	1.83	2.11	2.54	3.22	1.42	1.92	2.19	2.51	2.95
85–89	1.54	1.98	2.22	2.64	3.18	1.53	1.98	2.18	2.55	3.33
90–95	1.45	1.85	2.23	2.37	3.18	1.59	1.94	1.98	2.25	2.72

## Data Availability

Data supporting the reported results are available upon reasonable request from S.-F.L.

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
