# Peer review of "Functional Fitness Norms of Community-Dwelling Older Adults in Southern Rural Taiwan: A Cross-Sectional Study"

_healthcare, 2024, doi:10.3390/healthcare12020213_

Round 1

Reviewer 1 Report

Comments and Suggestions for Authors

Dear authors,

your Topic about "Functional fitness norms of community-dwelling older adults in Southern Rural Taiwan" is very interesting to characterize the population of Taiwan. Your aim "to determine the age- and gender-specific normative fitness scores for a large representative population of community-  dwelling older adults in Southern rural Taiwan" was very specific in the entire manuscript.

In my oppinion, your manuscript has one thing to be improved, that is the discussion section. You can compare your results with similar data from other countries. If you do this i thing the manuscript will be better.

Congratulations for your work

Author Response

We thank reviewer for their helpful suggestions and constructive comments. We hope that our revisions and responses detailed below point-by-point will improve the manuscript sufficiently for you to recommend publication.Please see the attachment.

Reviewer 2 Report

Comments and Suggestions for Authors

Dear Authors,

Please see some suggestions to improve the manuscript.

In abstract try to add more values about the significant results.

In introduction:

Please, explore and add more information regarding the "Enjoy Active Aging for Longevity and Life (EAALL)"

Please focus and add a litle more information about the main differences between man and women in aging concerning to physical fitness

Add if is possible the hypotheses of the study

In methods

There was some institution that agree with the study?

In "2.4. Functional Fitness Measure" please the description of each test performed and and also the BMI

They were tested in 2013 and 2014, after 10 years you think that is some impact of the results?

Author Response

(The authors gave the same response as above.)

Reviewer 3 Report

Comments and Suggestions for Authors

I would like to thank you for the opportunity to review this article. This article denotes a great effort by the authors to perform it. The article covers a very interesting and current topic that I am sure in the future they will be able to delimit better. Nevertheless, in my opinion, some parts need to be improved. I detail the comments below:

ABSTRACT

P.1, line 21. You can also include the main statistical values of your results.

P.1, line 31. You can also include other keywords “Taiwan”, and “Physical Activity”

INTRODUCTION

The introduction section needs more references to support the affirmations, for example, the sentences of lines 37,42,43,46 o 47.

P.2, line 49. It seems to have a mistake in the reference format, please could you check it and rewrite to do it adequate to the journal format?

P.2, line 56. It would probably be enriching to provide more information on the rejection of the tests, but other aspects could be reduced. In general, in my opinion, the introduction section should be reduced.

P.2, line 91. It could be better to only specify the aim of the actual study. 

METHODS

I have some doubts about the inclusion of volunteer participants from a single region of South Taiwan could introduce a risk of bias to the study, perhaps it could be good to mention it as a possible limitation in the discussion section.

Has this study been approved by an ethics committee? Please, could you specify it in the manuscript?

P.3, line 134.  Please, reference the questionnaire PAR-Q.

P.3, line 144.  To describe the battery of tests could improve the quality of the study, facilitating, another researcher to reproduce the results.

DISCUSSION

P.11, line 334. Could you reference the sentence “Given that older individuals and men are at increased risk of developing 334 chronic diseases,”?

REFERENCES

The references format changed in 1-5 from 6 to 27, please, could you rewrite it to do it more adequately?

Comments on the Quality of English Language

.

Author Response

(The authors gave the same response as above.)
